# Study on Characteristics of the Light-Initiated High Explosive-Based Pulse Laser Initiation

**DOI:** 10.3390/ma15124100

**Published:** 2022-06-09

**Authors:** Dengwang Wang, Jinglun Li, Yunfeng Zhang, Hao Li, Sheng Wang

**Affiliations:** 1Department of Nuclear Science and Technology, Xi’an Jiaotong University, No. 28, Xianning West Road, Xi’an 710049, China; wdw21s@stu.xjtu.edu.cn (D.W.); lijinglun1998@stu.xjtu.edu.cn (J.L.); 2Northwest Institutes of Nuclear Technology, No. 28, Pingyu Road, Xi’an 710024, China; zhangyunfeng@nint.ac.cn (Y.Z.); lihao@nint.ac.cn (H.L.)

**Keywords:** pulse laser, light-initiated high explosive, silver acetylene silver nitrate, threshold, structural response

## Abstract

The silver acetylene silver nitrate loading technology of the light initiated high explosive, as one of important means to simulate the structural response of powerful pulsed X-ray, adopts the pulse laser initiation. It has advantages of improvement of practical control, heterogenous loading realization and simultaneous loading timeliness. In this paper, the physical and mathematical models of hot spot initiation and photochemical initiation of energetic materials under the action of laser are firstly established, and then the laser initiation mechanism of the light initiated high explosive is specifically analyzed, and the laser initiation experiment is conducted based on the optical adsorption property of the light initiated high explosive. From this study, the laser initiation thresholds of 193 nm, 266 nm, 532 nm, 1064 nm wavelengths are given, and they are 5.07 mJ/mm^2^, 6.77 mJ/mm^2^, 7.21 mJ/mm^2^ and 10.61 mJ/mm^2^, respectively, and the complete detonation process is verified by detonation velocity. This work technically supports the study of pulse laser initiation process, mechanism and explosion loading rule as well as the loading technology of the light initiated high explosive to simulate the structural response of X ray.

## 1. Introduction

The light-initiated high explosive (LIHE) is a practicable experimental simulation technology used to simulate the structural response of a powerful pulsed X-ray [1], which can effectively solve problems, such as the complex heterogeneity of the shell, small specific impulse simulation, and simultaneous loading of large planar array, and it is also the only simulation technology that can simulate the material response caused by X-ray and structural response simultaneously.

The light-initiated high explosive (LIHE) facility, at this time, is used primarily to investigate the structural response of complex test items, such as re-entry bodies/vehicles to shock-producing events [2,3]. Tests at a LIHE facility are high-fidelity tests, meaning that the test loading is delivered in the proper time frame, and applied over the entire test surface at the same time [4]. During a hostile encounter—such as a nuclear weapon detonated in space near a re-entry vehicle—hot, warm, and cold X-rays are produced. When cold X-rays deposit themselves in a thin layer on the asset’s surface, that material heats up nearly instantaneously and vaporizes, sending a shock wave into the structure. This can cause many types of problems with external materials and internal components. By knowing the effects of these events on systems and components, designers can take action to counter them [5,6,7,8,9,10]. Light-initiated high explosive (LIHE) is the only available and highly realistic experimental technique to simulate the structural response of an intense pulse X-ray, which can solve the problems of complex shell anisotropy, simulation of small specific impulse, and simultaneous loading of large array. In a word, LIHE is a loading mode that causes the shell structure to respond.

Before a LIHE test [11] is remotely sprayed, a high explosive is detonated by a flash of light at Sandia’s light-initiated high-explosive (LIHE) facility. The explosives are remotely formulated on site in an environmentally controlled spray booth [12]. LIHE uses the primary explosive silver acetylide silver nitrate (SASN), which is highly sensitive and can be initiated with a bright flash of light. SASN is so sensitive that it is not widely used, and is carefully controlled throughout its life cycle at LIHE. After the explosive is made, a remote-controlled robot arm holding a modified spray gun carefully sprays thin layers of explosive onto the surface of complex structural shapes. Depending upon the complexity of the shape, and the desired explosive characteristics required, the spray process can last up to 10 h. After spraying and conditioning, the test unit is remotely moved to a test cell in front of a light array of tungsten wires enclosed in quartz tubes.

Put forward first by Brish et al. in 1966 [13], the energetic material of laser initiation is used to detonate lead azide and PETN explosives in the Q-switching action mode of neodymium glass laser. Erick et al. [14], Oestmark et al. [15], and Sun et al. [16], establish laser initiation models to study the laser detonation mechanism. Feng et al. [17] consider that after the temperature rises to critical, due to absorption of laser energy, the energetic material decomposes to release heat, and such a heat release and the external laser act together to make the reaction stronger, and then an explosion is caused. This initiation process caused by the heat is an ignition phenomenon. Liau et al. [18] found the non-light-supported re-ignition phenomenon during initiation, and the secondary combustion or explosion phenomenon from thermochemical process of heat accumulation in a self-maintaining chemical reaction.

Ali et al. [19] detonate HMX and TATB explosives through a CO_2_ laser, and then obtain the relations between initiation time and laser power, energy, and initiation temperature. Östmark et al. [20] study the influence of laser wavelength on explosive initiation through continuous laser with a wavelength 9~11 μm. Aleksandrov et al. [21] experimentally study the PETN detonation under a changing laser pulse width (7.5~100 ns), and find that the laser initiation energy density increases up to 1.4 times in a direct proportion with the increase in laser pulse width. Aduev et al. [22] and Tarzhanov et al. [23] study the influence of different concentrations of Al, Ni–C, and Al–C particles (100–220 nm) on the PETN laser initiation characteristics. Aluker et al. [24] and Aduev et al. [25,26] use the combined action of neodymium glass laser (1060 nm) excitation and copper heating to detonate PETN, study the initiation mechanism of energetic material, and find that in the PETN laser initiation, a 1.17 eV laser photon firstly induces PETN molecule light excitation, and then the PETN molecule transitions to the 0.4 eV energy barrier through the heating process. Khokhlov et al. [27] experimentally study the LIHE loading and laser initiation, and also study the detonation and transformation of flatly loaded explosive, with a diameter of 40 mm and a thickness of 5 mm, and the LIHE with a low density (ρ ≈ 0.9 g/cm^3^) made of fine-particle RDX and aluminum. Zhang et al. [28,29,30,31] study the laser-sensitive energetic complex primary explosive, analyze the advantages and problems of all kinds of agents, partially summarize the laser initiation mechanism, and expect the research and development of future new laser-initiating explosives.

This paper tentatively explores the laser initiation mechanism, initiation thresholds, and spectral characteristics under different wavelengths for a LIHE used to simulate the structural response of a powerful pulsed X-ray, which technically supports the simultaneous loading of large-area laser initiation.

## 2. Interaction between Laser and Explosive

When the laser acts on the energetic material, there is usually thermal effect, impact effect, photochemical effect, ionization, breakdown, etc., between them, and the corresponding laser initiation mechanism can be divided into the thermal initiation mechanism, impact initiation mechanism, photochemical mechanism, etc. Currently, the laser initiation model established by Sun et al. [16] is often used, which is shown in Figure 1, and it is a typical thermal mechanical theory, based on the heat conduction. For the chemical reaction term, the typical laser initiation model is shown as follows:(1)$$\rho c\frac{\partial T}{\partial t}=K\frac{\partial^{2}T}{\partial x^{2}}+aI_{0}+\rho QA\exp(-\frac{E}{RT})$$
where *ρ* is the density of agent; *C* is the thermal capacity of agent 1; *K* is the heat conductivity coefficient of agent; *I*_0_ is the laser intensity; *Q* is the chemical reaction heat of agent; *A* is the frequency factor; and *E* is the activation energy of agent.

### 2.1. Thermal Mechanism Process Analysis

To study the ignition characteristics of LIHE under the action of laser, the laser should first be expressed in a form of heat, and then conducted to LIHE. The distribution rule equation of internal temperature of a heat-conducting material is a differential equation of heat conduction. The expression under a rectangular coordinate system is shown as follows:(2)$$\rho c\frac{\partial T}{\partial t}=\frac{\partial }{\partial x}(\lambda \frac{\partial T}{\partial x})+\frac{\partial }{\partial y}(\lambda \frac{\partial T}{\partial y})+\frac{\partial }{\partial z}(\lambda \frac{\partial T}{\partial z})+\phi$$

The heat conduction equation without internal heat source meets Laplace’s equation:(3)$$\frac{\partial^{2}T}{\partial x^{2}}+\frac{\partial^{2}T}{\partial y^{2}}+\frac{\partial^{2}T}{\partial z^{2}}=0$$

For uniform LIHE, it is assumed that its chemical and physical properties remain unchanged during temperature rising, and it is known from the heat conduction theory that the energy conservation equation of ignition heat is shown as follows:(4)$$\rho c\frac{\partial T}{\partial t}=\lambda (\frac{\partial^{2}T}{\partial x^{2}}+\frac{j}{x}\frac{\partial T}{\partial t})+\rho QA\exp(-\frac{E_{a}}{RT})+(1-f)\alpha I\exp(-\alpha b)$$
where (1−f)αIexp(−αx) is laser energy; $\rho QA\exp(-\frac{E_{a}}{RT})$ is chemical reaction heat; and *j* = 0, 1 represents 1D and 2D models, respectively. The boundary condition meets category II boundary conditions, and it is expressed as follows:(5)$$\rho c\frac{\partial T}{\partial t}=\lambda \frac{\partial^{2}T}{\partial x^{2}}+\rho QA\exp(-\frac{E_{a}}{RT})\left(0\leq x\leq a_{0}\right)-\frac{dT}{dx}=0\left(x=0\right)$$
where ρ is the density of LIHE (kg/m^3^); *c* is the specific heat of LIHE (J/kg·K); λ is the heat conductivity coefficient (W/m·K); *Q* is the chemical reaction heat of LIHE (J/kg); *A* is the frequency factor (S^−1^); *R* is the universal gas constant (J/mol·K); *I* is the incoming laser intensity (W/cm^2^); f is the reflectivity of agent; *α* is the absorption coefficient of the sample to the laser (1/m); b is the convective heat transfer coefficient of sample and environment; *E_a_* is the activation energy of agent (J/mol); and *a*_0_ is the thickness of the agent. It is assumed that the laser distribution is uniform, so
(6)$$P(t)=\frac{P[H(t)-H(t_{1})]}{\pi \omega_{r}^{2}}$$
where H(t) is Heaviside function, which means that H(t)=1, *t* > 0; H(t)=0, *t* ≤ 0; and *ω**_r_* is the laser beam radius. With Laplace conversion and convolution theorem application, the analytical solution of energy conservation equation is shown as follows
(7)$$\begin{array}{l} T(x,t)=T_{0}+\frac{(1-f)\eta P\sqrt{a}}{\pi \omega_{r}^{2}\lambda }[\frac{2\sqrt{t}}{\sqrt{\pi }}\exp(-\frac{x^{2}}{4at})-\frac{x}{\sqrt{a}}erfc(\frac{x}{2\sqrt{at}})]H(t)\\ -[\frac{2\sqrt{t_{1}}}{\sqrt{\pi }}\exp(-\frac{x^{2}}{4att_{0}})-\frac{x}{\sqrt{a}}erfc(\frac{x}{2\sqrt{at_{0}}})]H(t_{0}) \end{array}$$
where *η* is the photothermal conversion coefficient and *erfc* is the error function; then
(8)$$erfc(x)=\frac{2}{\sqrt{x}}\int_{0}^{x}e^{-t^{2}}dt$$

The expression of *T* on the surface of energetic material (*x* = 0) is shown as follows:(9)$$T(0,t)=T_{0}+\frac{2(1-f)\eta \sqrt{a}}{\lambda \pi^{3/2}\omega_{r}^{2}}[\sqrt{t}H(t)-\sqrt{t_{0}}H(t_{0})]$$
(10)$$\Delta T=\frac{2(1-f)\eta }{\pi^{3/2}\omega_{r}^{2}}\sqrt{\frac{t}{\rho c\lambda }}H(t)$$

The ignition energy of energetic material is shown as follows:(11)$$E=\frac{\pi^{3}\omega_{r}{}^{4}\rho c\lambda {\left[T_{1}-T_{0}\right]}^{2}}{4\alpha^{2}\eta 2PH{\left(t_{1}\right)}^{2}}$$

With the laser ignition process based on thermal mechanism analyzed, the physical model and the mathematical model are established. Upon the calculation of the heat conduction equation, established by boundary conditions, the relations between the critical energy of the ignition and the initial parameter of energetic material, and the relations between the material surface and the time, can be obtained. It can be seen from Equation (9) that the rise in surface temperature of the energetic material is directly proportional to the laser power, and the energy of the laser ignition is inversely proportional to the power. From the results, it is known that the increase in laser power positively affects the laser ignition process, based on the thermal mechanism. Moreover, as the energy output of the laser remains unchanged, it is of significant importance to reduce the loss of laser during transmission, for the improvement of ignition performance.

### 2.2. Photochemical Mechanism Process Analysis

The photochemical effect of LIHE is analyzed based on its optical absorption property. A SASN molecule absorbs the laser photon with a specific frequency and then dissociates, and the high-activity fast particles from the dissociation lead to a further chemical chain reaction, so that the ignition is caused; this is a photochemical ignition. Under the action of a specific frequency laser, the material molecule directly photolyzes, and causes the chain reaction in the material. The explosion reaction of Ag_2_C_2_·AgNO_3_ is shown as follows:Ag_2_C_2_·AgNO_3_→3Ag + CO_2_ + CO + N_2_(12)

For example, when SASN is radiated by a laser with a wavelength 190 nm, the reaction is shown as follows:Ag_2_C_2_ + hv (190 nm)→2Ag + 2C(13)
2C + 2AgNO_3_→2CO + 2Ag + 2O_2_ + N_2_(14)
2CO + O_2_→2CO_2_(15)

The final result when a wavelength is 190 nm is:Ag_2_C_2_·AgNO_3_ + hv→3Ag + CO_2_ + CO + N_2_(16)

The above reaction process is caused by the photochemistry, and contains the chain mechanism. The dissociation caused when a SASN substance molecule absorbs several photons simultaneously or successively is called the multiple photon dissociation. The transition probability in the n photon transition process is shown as follows
Wn = ∑_n_n_0_^n^ + O(n_0_^n^)(17)
where ∑n (unit cm^2n^ s^n−1^) is n step transition section, n_0_ is the photon stream density, and Wn is n photon transition probability. Without the intermediate state resonance, item 2 in the Equation (8) can be ignored, and then:Wn = ∑_n_n_0_^n^ = ∑_n_(I_0_/hv)^n^(18)

The above equation means that the transition probability Wn of n photon absorption is directly proportional to the nth power of the laser intensity I_0_. Usually, the single-photon absorption section ∑_1_ ranges from 10^−16^ to 10^−22^ cm^2^, while the double-photon absorption section ∑_2_ ranges from 10^−48^ to 10^−57^ cm^4^ s, and the ∑n value greatly decreases with the increase in n. Therefore, the multiple-photon phenomenon is rarely observed under normal light source conditions. As the laser power density I_0_ in the Equation (19) is large enough (I_0_ ≥ 10^6^ W/cm^2^), there is an obvious multi-photon absorption (MPA) phenomenon, so that the observable multi-photon dissociation (MPD) effect will occur. Two conditions need to be met for SASN initiation under the photochemical action of laser:The laser wavelength is strictly matched with the absorption wavelength of SASN, and then the energetic material can be photolyzed, due to resonance absorption to the laser;The laser energy of irradiation SASN is not too small.

## 3. Pulse Laser Initiation Platform

The experimental system of laser initiation platform consists of a laser, light path, SASN sample, energy meter, spectrometer, photoelectric probe, high-speed camera, and experimental protector. The schematic diagram of system composition and real platform are shown in Figure 2 and Figure 3. The Photocell is ET-2030 of ETL, the spectrometer is AvaSpec-ULS2048, and the spectral measurements range is 200 nm to 1400 nm. The pressure is CA-1135 of Dynasen, the high-speed camera is V2512 of Phantom.

Two laser initiation platforms were set up based on different experimental requirements. The laser included a ArF excimer laser and Q-smat450 pulsed laser. Table 1 shows the basic parameters of the laser. The ArF excimer laser had a main wavelength 193 nm ultraviolet light and obvious attenuation in the air, and the test results of laser attenuation in the air are shown in Table 2. It can be seen that LIHE is not conducive to practical application, although it has a high-absorption for ultraviolet light.

The 2D and 3D representations of the distributions of energy of the Qsmart-450 laser with different wavelengths are shown in Figure 4. From the figure, it is seen that the energy platform of laser obviously facilitates the synchronous loading of laser initiation LIHE within the beam spot range.

## 4. Properties of Silver Acetylene–Silver Nitrate of LIHE

The silver acetylene–silver nitrate is the complex of silver acetylene and silver nitrate, with a molecular formula Ag_2_C_2_·AgNO_3_, and is insoluble in water, ethanol, diethyl ether, and acetone. In the conventional synthesis and preparation method, the acetylene gas is injected into the aqueous solution of silver nitrate, and the white flocculent precipitate generated is Ag_2_C_2_·AgNO_3_.
(19)$$C_{2}H_{2}+3{\mathrm{AgNO}}_{3}\overset{H_{2}O}{\to }{\mathrm{Ag}}_{2}C_{2}\cdot {\mathrm{AgNO}}_{3}↓+2{\mathrm{HNO}}_{3}$$

Silver acetylene–silver nitrate is an unstable substance, which decomposes under a strong light exposure and generate a huge amount of gas, as well as gives out heat. The decomposition reaction equation is shown as follows:(20)$${\mathrm{Ag}}_{2}C_{2}\cdot {\mathrm{AgNO}}_{3}\to 3\mathrm{Ag}+{\mathrm{CO}}_{2}+\mathrm{CO}+\frac{1}{2}N_{2}+773\mathrm{kJ}$$

The surface topography of the sample under different resolutions is characterized with the scanning electron microscope (SEM), as shown in Figure 5. The sample is formed by spherical nanoparticles with a diameter of 90 nm, and there is neat and orderly crystal formation, a smooth surface, and uniform size distribution; all these properties facilitate the synchronous initiation and loading.

The X-ray diffraction (XRD) was used to characterize the test (scanning range: 5~90°), and the test results are shown in Figure 6. It shows that the position of sample diffraction peak is basically consistent with that of XRD diffraction peak of the silver acetylene–silver nitrate, and this verifies the composition of LIHE.

The sample was tested in infrared by a Fourier infrared spectrometer, and the wave number tested ranged from 4000 to 500 cm^−1^. The test results are shown in Figure 7. The functional group region is 3700~1333, while the fingerprint region is 1333~650, and the infrared spectrograms of the sample include two strong absorption peaks of CO_2_ in the atmosphere. They are near 2349 cm^−1^ and 667 cm^−1^, and meet the infrared spectral characteristics of the acetylene bond (-C≡C-) in the silver acetylene. For samples, the absorption peak appears at 1385 cm^−1^ or 840 cm^−1^, and they correspond to the antisymmetric stretching and vibration peaks of silver nitrate in the silver acetylene–silver nitrate.

The SASN full-band absorption spectrum curve is shown in Figure 8. It is seen that it has a good absorption of ultraviolet light, with a wavelength ranging from 190 nm to 300 nm, and the SASN absorption reduces quickly between 300~450 nm, while the SASN absorption is not obvious if the wavelength is greater than 450 nm.

## 5. Experimental Study on Laser Initiation

LIHE specimen is shown in Figure 9, and the specimen parameters are shown in Table 3. SASN was dripped in the center of the aluminum plate. A Dynasen pressure probe was placed in the center to measure the pressure and time of arrival. The surface density of SASN was 10–70 mg/cm^2^

### 5.1. Initiation Threshold

Table 4 shows the experimental parameters of initiation of different lasers, while Figure 10 shows the experimental thresholds of laser initiation SASN.

From the experimental results, it is seen that SASN is detonated reliably under different wavelengths of pulsed laser, and the complete detonation is realized only when the energy density is greater than 5.07 mJ/mm^2^ at 193 nm, 6.77 mJ/mm^2^ at 266 nm, 7.21 mJ/mm^2^ at 532 nm, and 10.61 mJ/mm^2^ at 1064 nm. This also verifies that SASN light absorption reduces with the wavelength (see Figure 10). However, the complete detonation is realized only when the energy density is greater than 29.04 mJ/mm^2^ at 355 nm, and there is a higher initiation threshold at 355 nm than at any other wavelength, which is inconsistent with the characteristic rule inertia of SASN light absorption. Upon analysis, this might be because of a change in the SASN detonation method. The initiation manner of ultraviolet lasers of 193 nm, 266 nm, and 355 nm is mainly dominated by the photochemical ignition, while that of green light 532 nm is mainly dominated by hot spot ignition. These need to be further verified by the physicochemical microanalysis and process spectrum analysis.

### 5.2. Detonation Velocity Measurement of Laser Initiation

To judge the complete detonation, instead of combustion detonation, of laser initiation SASN, its detonation velocity was measured, and from the results, it is seen that the detonation velocity is above 1.3 km/s, and it is deemed as detonated completely. Table 5 and Figure 11 show the detonation velocity and measurement system and waveform, respectively.

### 5.3. Spectral Analysis of Initiation Process

The radiation spectrum of the detonation process of laser initiation LIHE measured by the fiber optic spectrometer is shown below, and Figure 12 is the radiation spectrum from the explosive explosion stimulated by the ArF laser (193 nm), while Figure 13 is the radiation spectrum from the explosive explosion stimulated by the third harmonic generation of Q-smart laser (355 nm). The measured spectrum mainly includes continuous spectrum and characteristic line spectrum. The continuous spectrum is mainly produced by the high-temperature gray body radiation of the explosion, while the characteristic line spectrum is mainly the radiation spectrum generated by a certain element or molecule in the explosion field, under high-temperature conditions or a chemical reaction. Upon the preliminary judgment, the corresponding elements of the measured spectral line mainly include: Ag, Na, K, etc., and the corresponding wavelengths are located as follows: Ag: 520.9 nm and 546.5 nm; Na: 589 nm; and K: 766.5 nm and 769.3 nm. Moreover, the characteristic radiation spectrum near 499 nm is obtained in some experimental measurements, but the element or molecule corresponding to this spectrum remains unknown. The element content in the experiment does not depend on the spectral intensity, and such intensity only means that this spectrometer has a higher absorption for this element. To calibrate the relation between element content and intensity, this spectrometer needs to be further calibrated in a stricter manner.

From spectral results, all element spectrums in the chemical exothermic reaction of SASN initiation are not obtained, so it is difficult to analyze the SASN exothermic reaction process, and this is related to the open experimental environment and complicated gas elements. In future experiments, a closed experimental environment (optical fiber for laser transmission) can be considered, and the inert gases can fill a container.

## 6. Conclusions

In this paper, the laser initiation model and the photochemical effect process of SASN were established and analyzed. This was followed by the set-up of experimental platform of low-power laser initiation using SASN, and the power density threshold, detonation velocity, spectrum, and other characteristic parameters of laser initiation SASN under different wavelengths were obtained, which provided a technical reference for a chemical explosion method in simulation of structural response of a powerful pulsed X-ray.

With the laser ignition process based on thermal mechanism analyzed, the physical model and mathematical model are established. Upon the calculation of the heat conduction equation, established by boundary conditions, the relations between the critical energy of ignition and the initial parameter of energetic material, and the relations between the material surface and the time, are obtained. The rise in surface temperature of energetic material is directly proportional to the laser power, but the energy of the laser ignition is inversely proportional to the power. From the results, it is known that the increase in laser power positively affects the laser ignition process, based on the thermal mechanism. As the energy output of the laser remains unchanged, reducing the loss of laser during transmission is conducive to the improvement of ignition performance;The photochemical reaction process is analyzed, based on the optical absorption property of LIHE, and the photochemical initiation conditions are proposed;The detonator initiation platform is set up, and the laser energy distribution and ultraviolet laser attenuation characteristics are obtained, while the property of silver acetylene–silver nitrate is investigated, using a variety of detection methods;The SASN laser initiation is verified by lasers with different wavelengths of 193 nm, 266 nm, 355 nm, 532 nm, and 1064 nm, and SASN is detonated reliably with the low energy. Then, the initiation threshold and spectral characteristics of laser initiation with different parameters are obtained. The ultraviolet laser initiation process is mainly dominated by the photochemical effect, while the visible light and infrared initiation process is mainly dominated by the hot spot effect, which verifies the light absorption characteristic of SASN.

## Figures and Tables

**Figure 1 materials-15-04100-f001:**
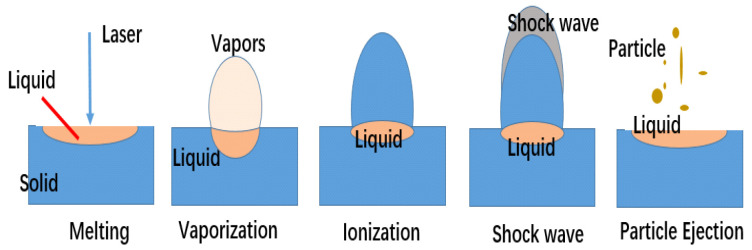
Schematic diagram of the thermal mechanism of laser initiation.

**Figure 2 materials-15-04100-f002:**
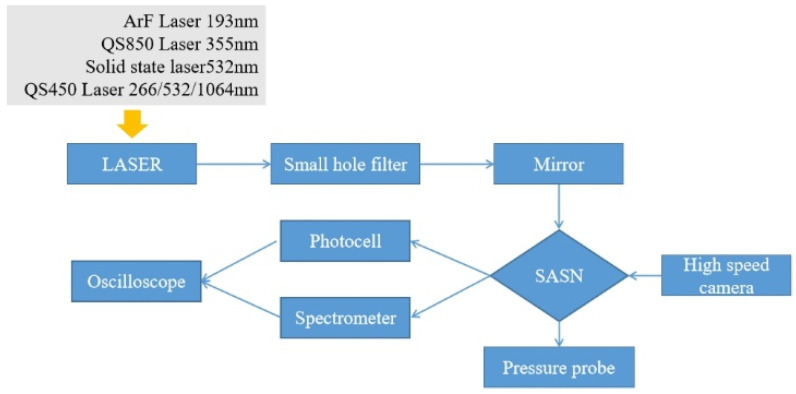
Schematic diagram of platform system composition.

**Figure 3 materials-15-04100-f003:**
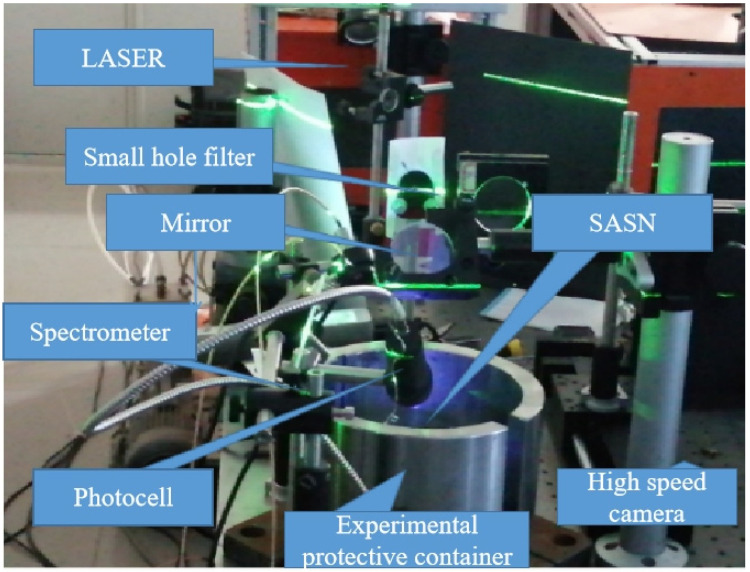
Different laser initiation platform systems.

**Figure 4 materials-15-04100-f004:**
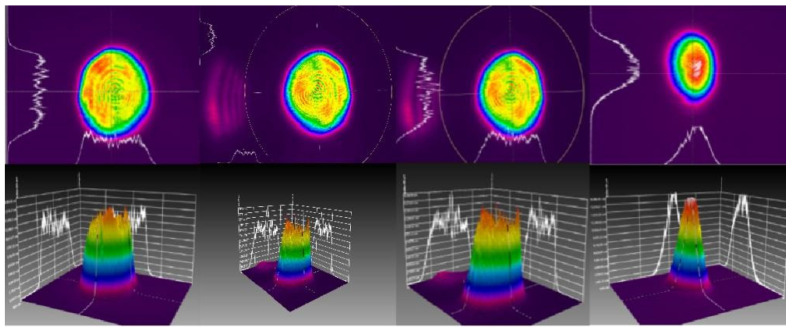
Energy distribution of laser at 266 nm, 355 nm, 532 nm and 1064 nm wavelength in 2D/3D.

**Figure 5 materials-15-04100-f005:**
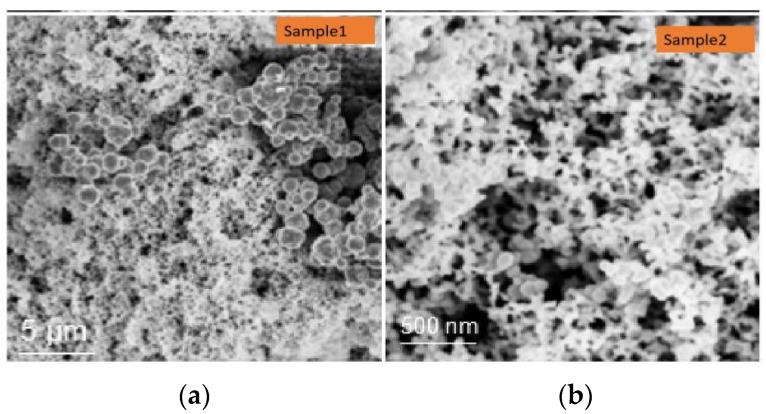
SEM of sample in different resolutions. (**a**) 5 μm. (**b**) 0.5 μm.

**Figure 6 materials-15-04100-f006:**
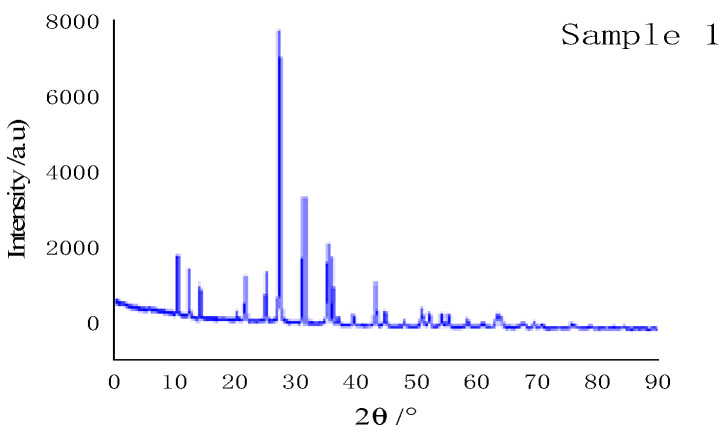
XRD pattern of sample.

**Figure 7 materials-15-04100-f007:**
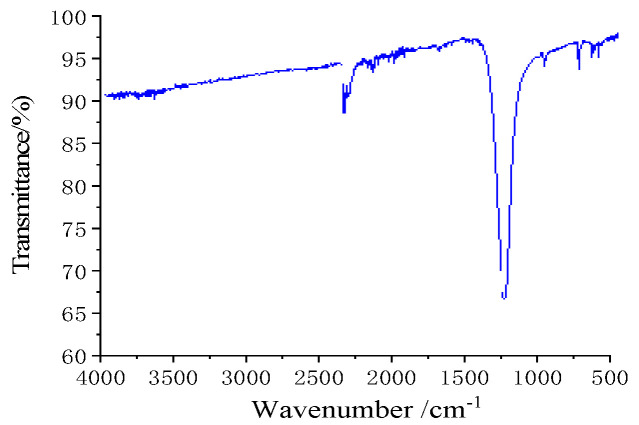
FT−IR pattern of sample.

**Figure 8 materials-15-04100-f008:**
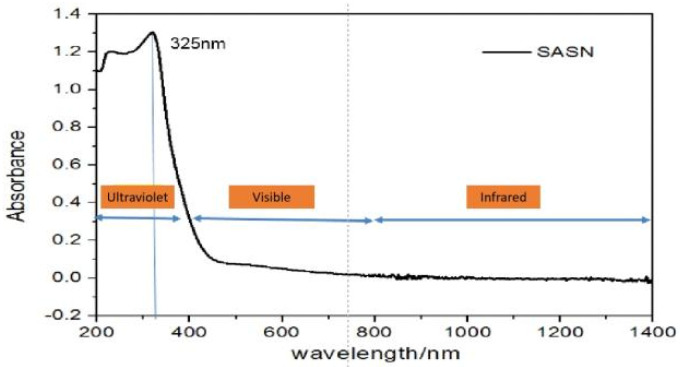
SASN full−band absorption spectrum curve.

**Figure 9 materials-15-04100-f009:**
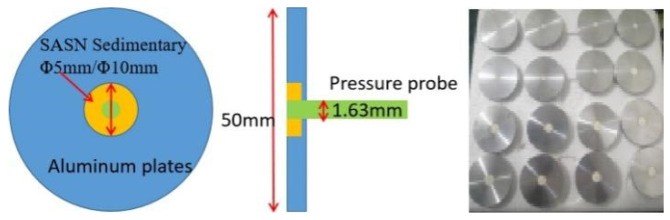
SASN specimen for ignition experiment.

**Figure 10 materials-15-04100-f010:**
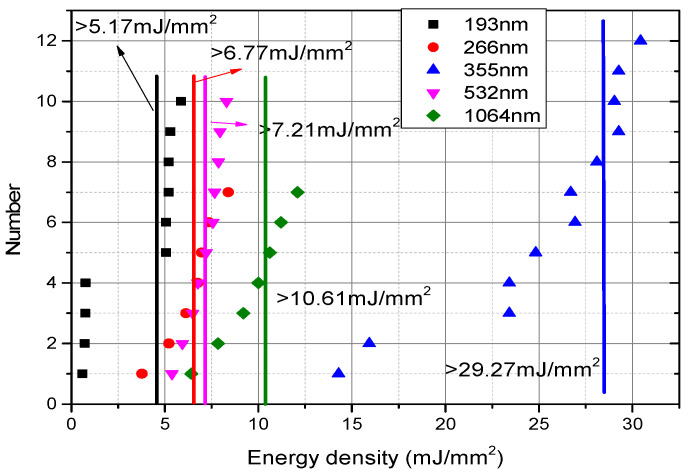
Experimental threshold of laser initiation SASN.

**Figure 11 materials-15-04100-f011:**
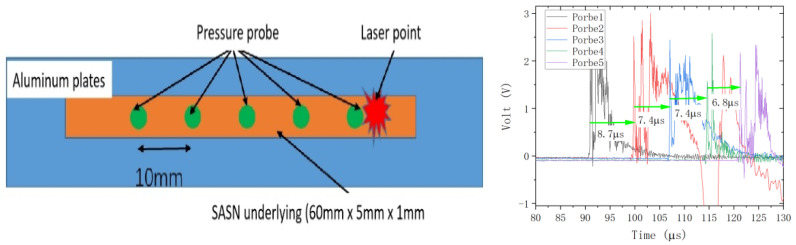
Measurement system and actually measured waveform.

**Figure 12 materials-15-04100-f012:**
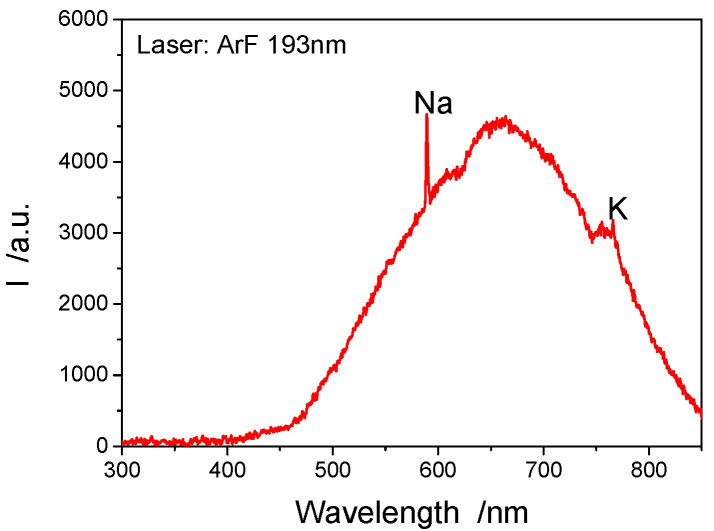
Experimental spectral data of SASN detonation by ArF 193 nm laser.

**Figure 13 materials-15-04100-f013:**
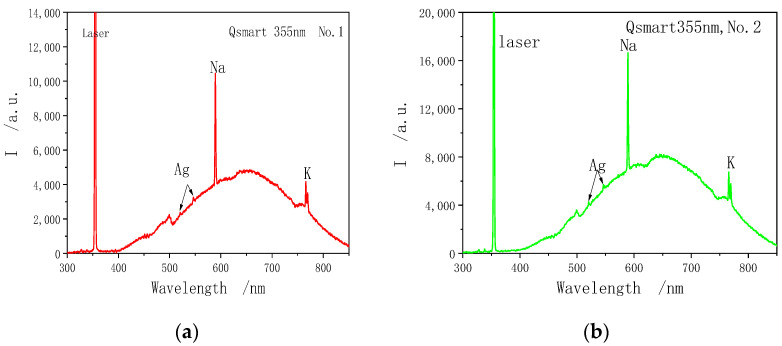
Experimental spectral data of SASN detonation by Q-Smart355 nm laser. (**a**) First experiment. (**b**) Second experiment.

**Table 1 materials-15-04100-t001:** Selected laser parameters.

Laser	Wavelength (nm)	Maximum of Single Pulse (mJ)	Pulse Width (ns)
ArF excimer laser	193	120	20
Q-smart450 Laser	266	50	6.3
	355	130	
	532	200	
	1064	400	

**Table 2 materials-15-04100-t002:** Air transmission attenuation test of ArF laser.

Input (mJ)	After Filter (mJ)	Output (mJ)	After Filter (mJ)
118	3.9	114	3.9
112	3.8	114	3.6
118	4.1	118	3.8
118	3.9	114	3.7

**Table 3 materials-15-04100-t003:** SASN specimen parameters.

Number	Plate Quality (g)	Total Quality (g)	SASN Quality (mg)	Area Density (mg/cm^2^)
1	19.1766	19.1830	6.4	32.59
2	18.3322	18.3427	10.5	53.48
3	18.4339	18.4384	4.5	20.92
4	18.0878	18.0931	5.3	26.99
5	17.8110	17.8229	11.9	60.61
6	18.9769	18.9807	3.8	19.35
7	18.5498	18.5537	3.9	19.86
8	17.9772	17.9820	4.8	24.45
9	18.2176	18.2306	27	42.45
10	19.1538	19.1660	12.2	19.18
11	18.9480	18.9594	11.4	17.92
12	19.5887	19.6018	13.1	20.60
13	18.2005	18.2149	14.4	22.64
14	18.8201	18.8349	14.8	23.27
15	18.8613	18.8720	10.7	16.82
16	18.6746	18.6932	18.6	29.25

**Table 4 materials-15-04100-t004:** Experimental parameter of initiation of different lasers.

Laser	Energy (mJ)	Light Spot (mm^2^)	Energy Density (mJ/mm^2^)	Power Density (kW/mm^2^)	Detonation
ArF, 20 Hz, 193 nm	3.1	5.275	0.59	29.5	No
	3.8	5.275	0.72	36	No
	4	5.275	0.75	37.5	No
	4	5.275	0.76	38	No
	3.8	0.75	5.07	253.5	Yes
	3.8	0.75	5.07	253.5	Yes
	3.9	0.75	5.2	260	Yes
	3.9	0.75	5.2	260	Yes
	4	0.75	5.3	265	Yes
	4.4	0.75	5.87	293.5	Yes
	13	0.427	30.44	5073.3	Yes
	12	0.427	7.86	1310.0	Yes
Q-smart450, 20 Hz, 266 nm	9.5	2.512	3.78	600.3	No
	13.1		5.21	827.8	No
	15.4		6.13	973.1	No
	17.0		6.77	1074.2	Yes
	17.5		6.97	1105.8	Yes
	18.5		7.36	1169.0	Yes
	21.1		8.40	1333.3	Yes
Q-smart450, 20 Hz, 355 nm	6.8	0.427	15.93	2653.9	No
	10		23.42	3901.8	No
	10.6		24.82	4135.0	No
	11.5		26.93	4486.5	No
	12		28.10	4681.5	No
	12.5		29.27	4876.4	Yes
	13		30.44	5071.3	Yes
Q-smart450, 20 Hz, 532 nm	16.8	3.120	5.38	854.7	No
	18.5		5.93	941.2	No
	21.1		6.76	1073.5	No
	22.5		7.21	1144.7	Yes
	23.9		7.66	1215.9	Yes
	24.8		7.95	1261.7	Yes
	25.9		8.30	1317.7	Yes
Q-smart450, 20 Hz, 1064 nm	22.6	3.524	6.41	1018.0	No
	27.6		7.83	1243.2	No
	32.4		9.19	1459.4	No
	35.2		9.99	1585.5	No
	37.4		10.61	1684.6	Yes
	39.5		11.21	1779.2	Yes
	42.6		12.09	1918.8	Yes

**Table 5 materials-15-04100-t005:** Detonation velocity measurement results.

Probe 1-Probe2 (μs)	Probe 2-Probe 3 (μs)	Probe 3-Probe 4 (μs)	Probe 4-Probe 5 (μs)	Average Velocity (km/s)
	7.1	6.8	7.1	1.429
	7.4	6.9	7.4	1.382
8.7	7.4	7.4	6.7	1.325
6.4	7.1	6.9	7.0	1.460

## Data Availability

All data included in this study are available upon request by contact with the corresponding author.

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
