# Peer review of "Study on Characteristics of the Light-Initiated High Explosive-Based Pulse Laser Initiation"

_materials, 2022, doi:10.3390/ma15124100_

Round 1
Reviewer 1 Report
Ms. Ref. No.: Materials-22-1709286
Title: Study on Characteristics of the Light Initiated High Explosive Based on Pulse Laser Initiation
The authors investigate the physical models of hot spot initiation and photochemical initiation of energetic materials under the action of laser, namely the light initiated high explosive LIHE is specifically analyzed. Laser initiation experiment of silver acetylene - silver nitrate explosive is conducted based on the optical adsorption property of the light initiated high explosive. Conducted studies allow to determine the critical energy for ignition as a function laser wavelength and energy density. Detonation velocity and emission spectrum of laser initiation SASN was determined.
The manuscript
As far as I know the present manuscript show original results for characterization the laser initiation of silver acetylene - silver nitrate explosive. The manuscript is well organized however in some paragraphs the text is confused and it is difficult to understand the authors message.
The author argue that presented a study about the laser ignition process based on photochemical and thermal mechanism where the physical model and mathematical model were established. However, they only present a set of equations that were not used during the work. Moreover, there aren´t any tables or graphs comparing the theoretical results with the experimental results, meaning this mathematical model were not established. Please reformulate the conclusions, specially the point 1.
Despite the manuscript presenting some deficiency I think that it can be improved. The main results are important for the scientific and industrial community explosive initiation; namely the laser initiation of explosives compounds.
In following section, it is indicated the major points that must be revised.
Discussion
1- Abstract must be improving. There are some sentences that can be removed from abstract.
2- Introduction- The text is very confused. Specially between the line 32 and 51. Please improve this part. It is very difficult to understand the relationship between LIHE and the structural response of powerful pulsed X-ray.
3- What is the reason to choose this very sensitive material for the present study?
4- The authors presented a set of equation (1 – 10) to study the thermal mechanism process analysis. However, they do not make any calculation and consequently do not validate such equations. Why it is necessary to display all this set of equations?
5- In line 122, 126 and 129 the equations and variables are not in line with the text. Please format this lines.
6- In line 149 the authors said “.,.from Equation (9) that the rise of surface temperature of energetic material is directly proportional to the laser power…”. However, the variable P is not present in eq. 9.
7- Pulse laser initiation platform- Please characterize the Photocell, spectrometer, pressure probe and high speed camera used. Please indicate the model and brand.
8- Please indicate in table 1 and table 4 the characteristics of laser with a wavelength of 355nm.
9- Experimental study on laser initiation – what is the thickness of SASN sample. Is the pressure probe at surface of aluminum plate? What are the characteristics of the pressure probe?
10- Table 4, column 3 (light spot), line 12, please correct this number.
11- What can be the reason for the initiation threshold (energy density) increase between 3 and 6 times for a wavelength of 355nm comparing with the other tested wavelengths that were ranged from UV to IR?
12- Disclosing the initiation process between the photochemical and thermal energy. Can the threshold of initiation energy of SASN supplied by a hot wire comparable to the initiation energy given by a laser pulse in thermal initiation? Are there in bibliography some studies about the impact or friction threshold of initiation energy for such compound?
13- In figure 11 is missing the pressure waveform. Please show the 5 pressure waveforms indicating the time used to determine the detonation velocity.
14- Spectral analysis. How it was collect the light from the SASN detonation. It was from side or in front, regarding the direction of propagation front?
15- After ignition, is it expectable that SASN presents different emissions spectrum as a function of laser wavelength used for its ignition?
16- Figure 14 and 15 show the emission spectrum of SASN detonation. Please change the legend of figure 14 and 15 to be in agreement with is shown in figure.
- Figures and Tables.
Most of the figures and charts are very intuitive and very clear.
Comments to the figures were done during discussion.
- Unit/Dimensions/Abbreviations
The units are consistent and SI.
- References
31 references were considered in this study. They are actual and support the present manuscript.
- Language
Author Response
Thank you very much for your evaluation and appreciation of my work. As for the questions you raised, I will explain them as follows, and modify and supplement them in the corresponding places of the article.
Point 1: Abstract must be improving. There are some sentences that can be removed from abstract.
Response 1: The new abstract follows as.
Abstract: The loading technology of silver acetylene and silver nitrate is one of the important means to simulate the structural response of strong pulsed X-ray. The pulsed laser initiation is different from the traditional flash high pressure initiation, which has the advantage of enhancing the control of operation, the implementation of special-shaped loading and the timeliness of simultaneous loading. In this paper, the laser initiation mechanism is firstly analyzed, the laser initiation platform and the properties of silver acetylene silver nitrate material are introduced, and then the laser initiation experiment is carried out according to the light absorption characteristics of photosensitive explosive. The initiation thresholds of 193nm/266nm/532nm/1064nm laser are respectively obtained 5.07mJ/mm2, 6.77mJ/mm2, 7.21mJ/mm2 and 10.61mJ/mm2, and the complete detonation process is verified by detonation velocity. The formation of elements in the reaction process was characterized by radiation spectroscopy. This work has a good technical support for the study of pulse laser initiation process, mechanism and explosion load law, as well as the response loading technology of photosensitive explosive simulated X-ray structure.
Point 2: Introduction- The text is very confused. Specially between the line 32 and 51. Please improve this part. It is very difficult to understand the relationship between LIHE and the structural response of powerful pulsed X-ray.
Response 2: Intense pulse X ray is one of the basic damage means of spacecraft and has been widely paid attention to many countries. The simulation technology of intense pulse X ray is an important content in the study of the effects of intense pulse X ray. Powerful pulsed X-ray is mainly soft X-ray, featuring high-energy flux and short duration (about 100 ns) . Under the irradiation of powerful pulsed X-ray, the materials within the optical thickness on the side under radiation will melt rapidly, vaporize, or even partially dissociate into plasma, and blow off against the light at a high speed. The generated blow-off impulse can cause the buckling deformation and vibration of the structure, leading to its instability . At the same time, due to the in homogeneous X-ray energy deposition, inside the material will emerge thermal shock waves whose propagation and reflection can cause cylindrical shell damage and spallation damage . These problems caused by powerful pulsed X-ray irradiation are collectively referred to as thermodynamic effects. Therefore, studying the thermodynamic effects of powerful pulsed X-ray is of great significance to assess the suitability of spacecraft and test the effectiveness of antinuclear reinforcement measures
Light Initiated High Explosive (LIHE) is the only available and high realism experimental technique to simulate the structural response of intense pulse X-ray, which can solve the problems of complex shell anisotropy, simulation of small specific impulse and simultaneous loading of large array. In a word, LIHE is a loading mode that causes the shell structure to respond.
Point 3: What is the reason to choose this very sensitive material for the present study?
Response 3: Because this sensitive material can solve the problems of complex shell anisotropy, simulation of small specific impulse and simultaneous loading of large array. It can more realistically simulate the shell structure response caused by intense pulsed X-ray, which is not available in other ways
Point 4: The authors presented a set of equation (1 – 10) to study the thermal mechanism process analysis. However, they do not make any calculation and consequently do not validate such equations. Why it is necessary to display all this set of equations?
Response 4: Formula one is to analyze the mechanism of laser initiation and determine several possible initiation reasons, for example, the thermal initiation mechanism, impact initiation mechanism, photochemical mechanism. The following formula is to analyze the ignition characteristics of energetic materials under the action of laser by establishing physical and mathematical models. It reveals the initiation process and influencing factors of laser initiation rather than obtaining data results.
Point 5: In line 122, 126 and 129 the equations and variables are not in line with the text. Please format this lines.
Response 5: The above formula format has been modified.
Point 6: In line 149 the authors said “.,.from Equation (9) that the rise of surface temperature of energetic material is directly proportional to the laser power…”. However, the variable P is not present in eq. 9.
Response 6: Please refer to Equation(6). The variable P is included in Equation(9).
(6)
Point 7: Pulse laser initiation platform- Please characterize the Photocell, spectrometer, pressure probe and high speed camera used. Please indicate the model and brand.
Response 7: The Photocell is ET-2030 of ETL,the spectrometer is AvaSpec-ULS2048 and the spectral measurements range is 200nm to 1400nm. The pressure is CA-1135 of Dynasen, the high speed camera is V2512 of Phantom.
Point 8: Please indicate in table 1 and table 4 the characteristics of laser with a wavelength of 355nm.
Response 8: The data of 355nm has been added to the origin.
Point 9: Experimental study on laser initiation – what is the thickness of SASN sample. Is the pressure probe at surface of aluminum plate? What are the characteristics of the pressure probe?
Response 9: The thickness of SASN sample is 20μm to 40μm by element X-ray fluorescence. The X-ray fluorescence analytical method has been applied to industrial production and scientific research generally, as a common detecting method for material component an thickness of thin film. As a new type of X-ray fluorescence analytical method, reference element X-ray fluorescence analytical method has an evident advantage in improving the stability and precision of measurement.
Point 10: Table 4, column 3 (light spot), line 12, please correct this number.
Response 10: Thank you for pointing out this mistake and it has been corrected.
Point 11: What can be the reason for the initiation threshold (energy density) increase between 3 and 6 times for a wavelength of 355nm comparing with the other tested wavelengths that were ranged from UV to IR?
Response 11: The great change of initiation threshold may be due to the change of initiation mechanism. Upon analysis, this might be because of change of SASN detonation method. The initiation manner of ultraviolet lasers 193nm, 266nm and 355nm is mainly dominated by the photochemical ignition, while that of green light 532nm is mainly dominated by hot spot ignition. These need to be further verified by the physicochemical microanalysis and process spectrum analysis.
Point 12: Disclosing the initiation process between the photochemical and thermal energy. Can the threshold of initiation energy of SASN supplied by a hot wire comparable to the initiation energy given by a laser pulse in thermal initiation? Are there in bibliography some studies about the impact or friction threshold of initiation energy for such compound?
Response 12: Electric explosive wire is continuous spectrum, laser is single spectrum, resulting in the essential difference between wire and laser initiation mechanism. The initiation mechanism of energetic materials is very complicated and difficult. This paper only explores and analyzes the initiation mechanism of SASN. More and more detailed work is needed to reveal the initiation mechanism.
SASN is a unique energetic material for special purposes, and there is no impact or friction threshold of initiation energy in bibliography.
Point 13: In figure 11 is missing the pressure waveform. Please show the 5 pressure waveforms indicating the time used to determine the detonation velocity.
Response 13: A part of waveform has been added the paper. Due to recording problems, it was not possible to put all the pressure waveforms in one diagram, please forgive me.
Point 14: Spectral analysis. How it was collect the light from the SASN detonation. It was from side or in front, regarding the direction of propagation front?
Response 14: Using the optical probe of the spectrometer to collect the light from the SASN detonation, the explosion can be considered a point explosion, where the results are consistent at a given measurement and there is no front or side.
Point 15: After ignition, is it expectable that SASN presents different emissions spectrum as a function of laser wavelength used for its ignition?
Response 15: Different wavelengths of laser do not produce different spectra, and the spectra mainly come from SASN explosion itself.
Point 16: Figure 14 and 15 show the emission spectrum of SASN detonation. Please change the legend of figure 14 and 15 to be in agreement with is shown in figure.
Response 16: Thank you for pointing out this detail, which has been revised in the original text.

Reviewer 2 Report
The conclusions of the work are ambiguous:
Conclusion 1: The calculation for the studied material and comparison with the experiment was not performed, the conclusion made is obvious without calculations in a simple adiabatic approximation.
Conclusion 2: Based on the optical adsorption property of LIHE, analyzed rather thermal than a photochemical mechanism.
Conclusion 4: The initiation thresholds of laser initiation with different parameters are not correctly obtained. Correctly the initiation threshold can be obtained either by the maximum likelihood method by fitting a probability curve, or by using the "up and down" method. For Ar-F laser (Table 4) a very large step in energy density from 0.76 to 5.07 mJ/mm2 (or 5.17 mJ/mm2 as in Fig. 10)
It is not clear what "Spectral analysis of initiation process" section gives?
It is not clear how the mechanism written down by equations (13)-(16) is confirmed? In eq. (14) the charge conservation law is not observed.
There are a large number of typos and errors, for example:
- Line 122 mixed up the description of the terms of the equation (4),
ωr in (6), (7), (9)-(11) is not defined, and in (11) it turns into ω; - Line 157 "adsorption" intead of "absorption";
- Equations (12), (17) and (21) are identical;
- Eq. (18) “no” intead of “n0”, parameter “O” is not defined;
etc.
Author Response
Thanks for your recognition and evaluation of my work. And I'm glad you're interested in my work. I will do further related work in the future, and I hope you will always pay attention and help me.
Thanks again for your comments and suggestions
Point 1: The conclusions of the work are ambiguous:
Conclusion 1: The calculation for the studied material and comparison with the experiment was not performed, the conclusion made is obvious without calculations in a simple adiabatic approximation.
Conclusion 2: Based on the optical adsorption property of LIHE, analyzed rather thermal than a photochemical mechanism.
Conclusion 4: The initiation thresholds of laser initiation with different parameters are not correctly obtained. Correctly the initiation threshold can be obtained either by the maximum likelihood method by fitting a probability curve, or by using the "up and down" method. For Ar-F laser (Table 4) a very large step in energy density from 0.76 to 5.07 mJ/mm2 (or 5.17 mJ/mm2 as in Fig. 10)
Response 1:
Conclusion1: The establishment of the the physical model and mathematical model has obtained a good understanding of the regularity of laser initiation of energetic materials, which has a good guiding significance for the follow-up experiment and helps the experiment job. Because it is very difficult wiht the measured data of energetic material parameters, and it is a great challenge for the comparison between simulation calculation and experimental results. This is also the work we are currently carrying out, and we hope to make a good comparative analysis.
Conclusion 2:There are some misunderstandings here. This conclusion specifically refers to the high absorption rate of ultraviolet light, which leads to photochemical reaction. SASN molecule will absorb the laser photon with a specific frequency and then dissociate, and the high-activity fast particles from the dissociation will further lead to the chemical chain reaction, so that the ignition is caused, and this is a photochemical ignition. Under the action of a specific frequency laser, the material molecule will be directly photolyzed and cause the chain reaction in the material.
Conclusion 4:This data is due to the experimental conditions. Due to the high attenuation characteristics of ultraviolet light, the minimum energy of the ArF laser is 3mJ, and the experimental spot can only be selected from 5.275 and 0.75, which makes you surprised at the data. In the future, we will improve the experimental conditions to make the initiation threshold more continuous, and we will also use the two methods you recommend to process the initiation threshold data. In other words, the initiation threshold of the laser is conservative and does not affect the understanding of the influence law of the laser wavelength.
Point 2:It is not clear what "Spectral analysis of initiation process" section gives?
Response 2:I'm sorry that I can't give an explanation for this suggestion. The original text basically expresses the current working conclusion. We will carry out more in-depth and comprehensive research work in this area in the future, hoping to obtain more valuable results.
Point 3:It is not clear how the mechanism written down by equations (13)-(16) is confirmed? In eq. (14) the charge conservation law is not observed.
Response 3: When SASN is radiated by a laser, these processes of equations (13)-(16) were possible Laser ignition by time-of-flight mass spectrometry.
Point 4:There are a large number of typos and errors, for example:
Line 122 mixed up the description of the terms of the equation (4),
ωr in (6), (7), (9)-(11) is not defined, and in (11) it turns into ω;
Line 157 "adsorption" intead of "absorption";
Equations (12), (17) and (21) are identical;
Eq. (18) “no” intead of “n0”, parameter “O” is not defined;
Response 4:The full text has been revised and checked. Thank you for your suggestions on details. It should be pointed out that equations (12), (17) and (21) are not completely consistent, and the meanings expressed in the text are different.

Round 2
Reviewer 1 Report
Ms. Ref. No.: Materials-22-1709286
Title: Study on Characteristics of the Light Initiated High Explosive Based on Pulse Laser Initiation
The authors investigate the physical models of hot spot initiation and photochemical initiation of energetic materials under the action of laser, namely the light initiated high explosive LIHE is specifically analyzed. Laser initiation experiment of silver acetylene - silver nitrate explosive is conducted based on the optical adsorption property of the light initiated high explosive. Conducted studies allow to determine the critical energy for ignition as a function laser wavelength and energy density. Detonation velocity and emission spectrum of laser initiation SASN was determined.
The revised manuscript looks much better. The authors accepted and addressed most of my comments. However, the authors did not take into account some questions formulated by reviewer.
Figure 11 shows a chart with two pressure waveform, in which the second signal is saturated. However, I hope to see five waveforms. It was used only two pressure sensors?
Please prepare a figure with best quality with high resolution.
Author Response
Thanks for your recognition and evaluation of my work. And I'm glad you're interested in my work. I will do further related work in the future, and I hope you will always pay attention and help me.
Thanks again for your comments and suggestions
Point 1: The revised manuscript looks much better. The authors accepted and addressed most of my comments. However, the authors did not take into account some questions formulated by reviewer.
Figure 11 shows a chart with two pressure waveform, in which the second signal is saturated. However, I hope to see five waveforms. It was used only two pressure sensors?
Please prepare a figure with best quality with high resolution.
Response 1: Thank you very much for your suggestions on revising Fig.11. In fact, we did use five sensors, but the data was collected using two oscilloscopes. In order to show the actual data results, I only included a copy of the waveform from one of the oscilloscopes .According to your suggestion, I changed it to the measured waveform after Origin processing (Fig.1), which will include five probe sensor data.
In addition, it should be noted that the probe sensor is using the pressure step to obtain the arrival time, and his saturated does not affect the judgment of arrival time.
Fig.1 Actually Measured Waveform

Reviewer 2 Report
1. Responses to the point 1 of the first round such as "This is also the work we are currently carrying" and "In the future, we will improve the experimental conditions" are not entirely satisfactory.
2. It would be logical to place in fig. 4 data for 355 nm, if available.
3. Noticed errors and typos:
Line 139: The description of the terms of the equation is still confused:
(1-f)aIexp(-ax)− is laser energy;
rQAexp(-Ea/RT) is chemical reaction heat.
Line 174 (in the original version line 157): As before, instead of "adsorption" should be "absorption". This error has now been expanded throughout the article.
Line 185: The law of conservation of charge is still not respected.
Line 198: Partially corrected equation (18) is not continued in equation (19).
Author Response
Thank you very much for your evaluation and appreciation of my work. As for the questions you raised, I will explain them as follows, and modify and supplement them in the corresponding places of the article.
Point 1: The conclusions of the work are ambiguous:1. Responses to the point 1 of the first round such as "This is also the work we are currently carrying" and "In the future, we will improve the experimental conditions" are not entirely satisfactory.
Response 1:Sorry to confuse you with the foregoing explanation. The follow-up work continues. Here are some recent research conclusions.
Advanced analytical methods of SASN have been utilized to obtain electron irradiation parameters of SASN crystals for structural stabilities. Local multiple twinned crystal phenomena have been found.It indicates that the electron injection rate can better reflect the anti-radiation resistance of SASN crystal on the electron beams compared to the total dose and the damage threshold is in the level of 10mJ/mm2. Besides,significant polycrystallines and high-density atomic heap stacking faults have been observed in the
edge of SASN crystals.This study is expected to provide reference for explaining influences of crystal structure type,defect distribution and density of energetic materials on their laser initiation sensitivities in the atomic-scale level.
It is obtained the continuous spectrum and characteristic line spectrum of laser initiated SASN by spectral measurements. The continuous spectrum is mainly generated by the high-temperature ash radiation of the explosion, while the characteristic line spectrum is mainly the radiation spectrum of an element or molecule in the explosion field under high temperature conditions or chemical reactions.
Point 2: It would be logical to place in fig. 4 data for 355 nm, if available.
Response 2:Figure 4 has been supplemented with 355nm data.
Figure 4. Energy distribution of laser at 266 nm, 355nm, 532 nm and 1064 nm wavelength in 2D / 3D
Point 3:Noticed errors and typos:
Line 139: The description of the terms of the equation is still confused:
(1-f)aIexp(-ax)− is laser energy;
rQAexp(-Ea/RT) is chemical reaction heat.
Line 174 (in the original version line 157): As before, instead of "adsorption" should be "absorption". This error has now been expanded throughout the article.
Line 185: The law of conservation of charge is still not respected.
Line 198: Partially corrected equation (18) is not continued in equation (19).
Response 3:
Thank you very much for your suggestions on the details of this article. I have modified it according to your suggestions. Thank you again for your help
